# Changes of Exercise, Screen Time, Fast Food Consumption, Alcohol, and Cigarette Smoking during the COVID-19 Pandemic among Adults in the United States

**DOI:** 10.3390/nu13103359

**Published:** 2021-09-25

**Authors:** Liwei Chen, Jian Li, Tong Xia, Timothy A. Matthews, Tung-Sung Tseng, Lu Shi, Donglan Zhang, Zhuo Chen, Xuesong Han, Yan Li, Hongmei Li, Ming Wen, Dejun Su

**Affiliations:** 1Department of Epidemiology, Fielding School of Public Health, University of California Los Angeles, Los Angeles, CA 90095, USA; xiatong@g.ucla.edu; 2Department of Environmental Health Sciences, Fielding School of Public Health, University of California Los Angeles, Los Angeles, CA 90095, USA; jianli2019@ucla.edu (J.L.); tmatthews@ucla.edu (T.A.M.); 3School of Nursing, University of California Los Angeles, Los Angeles, CA 90095, USA; 4Department of Behavioral and Community Health Sciences, LSUHSC School of Public Health, 2020 Gravier Street, Room 213, New Orleans, LA 70112, USA; ttseng@lsuhsc.edu; 5Department of Public Health Sciences, Clemson University, Clemson, SC 29634, USA; lus@clemson.edu; 6Department of Health Policy and Management, College of Public Health, University of Georgia, Athens, GA 30602, USA; dzhang@uga.edu (D.Z.); zchen1@uga.edu (Z.C.); 7School of Economics, Faculty of Humanities and Social Science, University of Nottingham Ningbo China, Ningbo 315100, China; 8Surveillance and Health Services Research Program, American Cancer Society, Atlanta, GA 30303, USA; xuesong.han@cancer.org; 9Department of Population Health Science and Policy, Department of Obstetrics, Gynecology, and Reproductive Science, Icahn School of Medicine at Mount Sinai, New York, NY 10029, USA; yan.li1@mountsinai.org; 10Department of Media, Journalism and Film, Miami University, Oxford, OH 45056, USA; lih19@miamioh.edu; 11Department of Sociology, University of Utah, Salt Lake City, UT 84112, USA; ming.wen@soc.utah.edu; 12Department of Health Promotion, College of Public Health, University of Nebraska Medical Center, Omaha, NE 68198, USA; dejun.su@unmc.edu

**Keywords:** COVID-19, fast food, exercise, screen time, alcohol, smoking, lifestyles

## Abstract

Objective: To investigate the impact of the COVID-19 pandemic on multiple lifestyle changes among adults in the United States (USA). Methods: We conducted a survey, the Health, Ethnicity, and Pandemic (HEAP) Study, in October 2020 among USA adults. Participants were selected from the United States using 48 sampling strata, including age, race, ethnicity, education, and gender, and were asked to report five lifestyle behaviors (i.e., exercise time, screen time, fast-food meal consumption, alcohol drinking, and cigarette smoking) before and during the COVID-19 pandemic. The associations of sociodemographic factors with each lifestyle change were estimated using weighted multivariable logistic regression models. Results: All 2709 HEAP participants were included in this study. Compared to pre-pandemic, the time spent on exercise decreased (32.06 vs. 38.65 min/day; *p* < 0.001) and screen time increased (6.79 vs. 5.06 h/day; *p* < 0.001) during the pandemic. The percentage of individuals who reported consuming fast-food meals ≥3 times/week decreased from 37.7% before the pandemic to 33.3% during the pandemic. The percentage of heavy drinkers (≥5 times/week) increased from 20.9% before the pandemic to 25.7% during the pandemic. Among smokers, heavy smoking (≥11 cigarettes/day) increased from 5.8% before the pandemic to 7.9% during the pandemic. We also identified subgroups who were more vulnerable to adverse influences from the pandemic, including racial/ethnic minority groups and young adults. Conclusions: The COVID-19 pandemic had negative impacts on multiple lifestyle behaviors among Americans. Mitigating such negative impacts of COVID-19 requires effective interventions, particularly for some vulnerable subgroups.

## 1. Introduction

The novel coronavirus disease (COVID-19), induced by severe acute respiratory syndrome coronavirus 2 (SARS-CoV-2), has caused a worldwide pandemic since March 2020. SARS-CoV-2 is a single-stranded RNA virus and the third coronavirus ((i.e., since SARS 2003, Middle East Respiratory Syndrome (MERS) 2012)) that has caused severe diseases in human in the past two decades [1]. COVID-19 is known to spread primarily via respiratory droplets during close contact. The symptoms of COVID-19 mainly appear 2–14 days after exposure to the virus and can be ranged from mild to severe, including fatigue, headache, fever, dry cough, sore throat, new loss of taste and smell, muscle or body aches, diarrhea, difficulty breathing, sepsis, and acute respiratory failure [2]. Approximately 5% of patients who were infected with COVID-19 had severe symptoms and needed intensive care [3]. As of 6 September 2021, more than 220 million people have been infected by the SARS-CoV-2 and 4.5 million died worldwide [4].

In the 2003 Severe Acute Respiratory Syndrome (SARS) pandemic, several studies documented changes of health behaviors in people living in the most infected areas, such as China [5,6]. During the current COVID-19 pandemic, many countries, including the United States (USA), implemented measures and recommendations to restrict non-essential activities and reduce the spread of the disease. While these restrictions were helpful to reduce human-to-human infection of COVID-19, they could result in profound changes in people’s normal daily activities and behaviors, including travel, shopping, exercise, food consumption, and eating patterns, as well as smoking and alcohol drinking. For example, levels of daily physical activity and exercise might be significantly reduced due to the closures of gyms, recreational facilities, group exercise classes, and travel restrictions. On the other hand, working, meeting, and learning remotely are likely to increase screen time and sedentary behaviors. In addition, restaurant closures and increased online grocery shopping might also change food consumption and dietary behaviors.

Since the beginning of the COVID-19 pandemic, several studies from different countries have observed undesired lifestyle changes among their populations, such as less exercise, more sedentary behavior, unhealthy dietary patterns, and increased alcohol consumption and cigarette smoking in Canada [7], Italy [8,9], Brazil [10], and Poland [11], as well as an international study via online survey [12]. To the best of our knowledge, only one small-scale local study examined physical activity and diet during the pandemic using data from 112 desk workers in the USA [13]. Two other USA studies only targeted alcohol drinking or cigarette smoking [14,15].

The objective of our study was to investigate the impact of COVID-19 on multiple lifestyle behaviors in USA adults and to identify sub-groups who were more adversely impacted by the COVID-19 pandemic, using a nationally representative, population-based sample.

## 2. Methods

### 2.1. Study Design and Population

This analysis was conducted using the data from the Health, Ethnicity, and Pandemic (HEAP) Study. HEAP is survey designed by a consortium of investigators from several universities in the USA, led by the Center for Reducing Health Disparities at the University of Nebraska Medical Center. The HEAP survey was carried out by the National Opinion Research Center (NORC) at the University of Chicago with the sample randomly drawn from NORC’s AmeriSpeak Panel and Dynata panel. AmeriSpeak is a probability-based mixed-mode panel designed to be representative of the USA non-institutionalized population using a multi-stage address-based sample design. Participants were selected using 48 sampling strata, including age, race, education, and gender following the American Association for Public Opinion Research guidelines. Detailed information regarding the panel recruitment and selection methodology can be found in the previous publication [16]. Figure 1 provides the flowchart of participant selection in the AmeriSpeak panel. The AmeriSpeak panel was supplemented with respondents who were Asian-American from the Dynata panel. The HEAP survey was conducted between October and November 2020 and was delivered both in English and Spanish through the internet or telephone. Participants received $3 in cash or the equivalent as an incentive to complete the survey. NORC implemented stratified sampling and poststratification weighting procedures to ensure the study sample was nationally representative of USA adults (age ≥18 years). The study was approved by the Institutional Review Board at the NORC.

### 2.2. Survey Design and Measures

The HEAP is a cross-sectional study in design and the survey included questions related to mental health, lifestyle changes, racial discrimination, financial status, and healthcare utilization. Lifestyle changes were assessed using five questions asking participants to report the following behaviors before and during the pandemic: (1) number of minutes spent on exercise (e.g., running, walking, swimming, sports, yoga, and strength training) per day; (2) the number of fast-food meals (e.g., burgers, fries, and pizza) consumed per week; (3) the number of hours spent on screen time (e.g., TV, computer, cellphone, iPad, etc.) per day; (4) the number of cigarettes smoked per day; and (5) the number of alcoholic drinks consumed per week (one drink is equivalent to a 12-ounce beer, a 5-ounce glass of wine, or a drink with one shot of liquor). The survey also collected sociodemographic variables such as age, sex, race/ethnicity, educational attainment, marital status, annual household income, and health insurance coverage.

### 2.3. Statistical Analysis

All statistical analyses were performed with poststratification weighting to account for the complex survey design and sampling procedures. Each lifestyle behavior, before or during the pandemic, was reported as weighted mean (standard error, SE) for continuous variables or weighted percentage (actual frequency) for categorical variables. Differences in these behaviors during the pandemic were compared to behaviors before the pandemic using the Wilcoxon Signed Rank Test for continuous variables and Bhapkar’s Test for categorical variables. The prevalence of missing data ranged from 3.1 to 3.8% (including the response categories “Don’t know” and “Prefer not to answer”) for measures of lifestyle/behaviors. Changes in lifestyle behaviors were calculated (during pandemic–before pandemic) and categorized as “desired change” or “undesired change” for each behavior. The “undesired change” category refers to decreased exercise, increased screen time, increased numbers of fast-food meals, increased cigarettes smoking, or increased alcohol drinking during the pandemic as compared with before the pandemic, respectively, for each behavior. Associations of sociodemographic factors with changes in lifestyle behaviors were estimated using logistic regression models. Sociodemographic factors included as independent variables in the logistic regression models were age (18–29, 30–44, 45–59, vs. ≥60 years), sex (female vs. male), race/ethnicity (Non-Hispanic Blacks, Hispanics, Asians and Pacific Islanders (APIs), or Alaska Natives or Others vs. Non-Hispanic Whites), educational attainment (associate’s degree, bachelor’s degree, or higher vs. high school or less), marital status (widowed/divorced/separated or never married vs. married/living with a partner), annual household income ($25,000–49,999 or ≥$50,000 vs. <$25,000), and health insurance before the pandemic (uninsured, Medicare, or Medicaid vs. private insurance). All statistical analyses were performed with SAS statistical software (Version 9.4 SAS Institute Inc, Cary, NC, USA).

## 3. Results

### 3.1. Characteristics of Study Participants

All 2709 HEAP participants were included in this study. Among them, 20.5% were aged between 18–29 years, 25.5% between 30–44 years, 24.2% between 45–59 years, and 29.8% ≥ 60 years. The study population reflected the USA racial/ethnic composition, and the majority were Non-Hispanic Whites (61.3%), followed by Hispanics (16.7%), Non-Hispanic Blacks (11.9%), and Asian and Pacific Islanders (6.4%). Of all participants, 51.7% were female, 34.3% had a bachelor’s degree or higher, 58.2% were married or living with a partner, 58.6% had an annual household income ≥$50,000, and 52.7% had private health insurance (Table 1).

### 3.2. Lifestyle Behaviors before and during the COVID-19 Pandemic

Table 2 describes the distributions of five lifestyle behaviors before and during the COVID-19 pandemic. Compared with the pre-pandemic level, the daily average time spent on exercise decreased (32.06 vs. 38.65 min/day; *p* < 0.001), and screen time increased (6.79 vs. 5.06 h/day; *p* < 0.001) during the pandemic. Before the pandemic, 15.3% of individuals reported zero exercise time, and this proportion increased to 20.9% during the pandemic. The proportion of individuals who exercised ≥30 min per day also decreased from 56.4% before the pandemic to 45.4% during the pandemic. Individuals with screen time ≥4 h per day also increased from 59.5% pre-pandemic to 79.8% during the pandemic. For fast-food meals, the average consumption level was 1.41 times/week before the pandemic and decreased to 0.96 time/week during the pandemic. The percentage of zero consumption increased from 15.0% before the pandemic to 25.0% during the pandemic, and the percentage of consuming three or more fast-food meals per week decreased from 37.7% before the pandemic to 33.3% during the pandemic. The average alcohol drinking level among drinkers increased from 3.01 drinks/week before the pandemic to 4.24 drinks/week during the pandemic (*p* < 0.001). Non-alcoholic beverage drinking was unchanged during the pandemic (50.6% vs. 49.8%), but the percentage of heavy drinkers (≥5 drinks/week) increased from 20.9% before the pandemic to 25.7% during the pandemic. The average cigarette smoking level among smokers increased slightly from 9.29 times/week before the pandemic to 9.80 times/week during the pandemic (*p* < 0.001). The percentage of non-smokers was similar before and during the pandemic (82.9 vs. 82.4%). Quitting behavior was not statistically significant before and during the pandemic (only 0.2% of smokers became non-smokers). However, among smokers, heavy smoking (≥11 cigarettes/day) increased from 5.8% pre-pandemic to 7.9% during the pandemic. 

Figure 2 shows the percentage of participants who had undesired changes of five lifestyle behaviors (i.e., decreased exercise, increased screen time, increased numbers of fast-food meals, increased cigarettes smoking, or increased alcohol drinking during the pandemic as compared with before the pandemic). Overall, 31.2% of the participants experienced decreased exercise time, 60.4% increased screen time, 22.6% increased consumption of fast-food meals, 23.3% increased alcohol drinking, and 9.0% increased cigarette smoking during the pandemic compared to before the pandemic. 

### 3.3. Sociodemographic Factors Associated with Changes in Individual Lifestyle Behaviors

The relationship between sociodemographic factors and undesired changes in the five lifestyle behaviors are presented in Table 3. Compared to old adults aged 60 years or over, younger adults aged between 18–29 and 30–44 had higher odds of increasing consumption of fast-food meals and alcohol drinking during the pandemic; the age groups of 30–44 years (OR = 4.74; 95% CI: 1.80–12.48) and 45–59 (OR = 3.37; 95% CI: 1.28–8.85) years had higher odds of increasing smoking, and the age group of 30–44 years had higher odds (OR = 1.77; 95% CI: 1.13–2.77) of reduction in exercise time. Women had higher odds (OR = 1.72; 95% CI: 1.29–2.30) of increasing screen time than men. Compared to Non-Hispanic Whites, Non-Hispanic Blacks (OR = 1.64; 95% CI: 1.17–2.28) and Hispanics (OR = 2.30; 95% CI: 1.64–3.24) had higher odds of decreasing exercise time; Hispanics had higher odds of increasing screen time (OR = 1.92; 95% CI: 1.35–2.72); Non-Hispanic Blacks (OR = 1.49; 95% CI: 1.03–2.17), Hispanics (OR = 1.75; 95% CI: 1.21–2.52), and American Indian or others (OR = 3.65; 95% CI: 1.79–7.43) had higher odds of increasing consumption of fast-food meals. Non-Hispanic Blacks (OR = 1.45; 95% CI: 1.02–2.06) and Hispanics (OR = 1.59; 95% CI: 1.09–2.33) had higher odds of increasing alcohol drinking, while Asian Americans had lower odds of increasing alcohol drinking (OR = 0.51; 95% CI: 0.35–0.74) and increasing smoking (OR = 0.53; 95% CI: 0.29–0.97). Compared to individuals with high school education or less, those with Associate’s degrees (OR = 1.46; 95% CI: 1.03–2.08) or a bachelor’s degree or higher (OR = 1.79; 95% CI: 1.20–2.67) had higher odds of decreasing exercise time, and those with a bachelor’s degree or higher had greater odds of increasing screen time (OR = 1.90; 95% CI: 1.27–2.85) and alcohol drinking (OR = 1.78; 95% CI: 1.11–2.87). Individuals who never married had higher odds of decreasing exercise time (OR = 1.43; 95% CI: 1.02–2.01) than married individuals. Individuals with household income between $25,000 and $49,999 (OR = 0.37; 95% CI: 0.21–0.67) and >$50,000 (OR = 0.22; 95% CI: 0.11–0.43) had lower odds of increasing cigarette smoking during the pandemic compared to those who had annual household income <$25,000. Compared to individuals with private health insurance, those with Medicare had higher odds of decreasing exercise time (OR = 1.63; 95% CI: 1.06–2.52) and those with Medicaid had higher odds of increasing cigarette smoking (OR = 2.30; 95% CI: 1.03–5.17). 

## 4. Discussion

In this large study among USA adults with multiple racial/ethnic groups, we found that the COVID-19 pandemic resulted in undesired changes in multiple health-related lifestyle behaviors. On average, USA adults decreased their time spent on exercises, but increased their screen time during the COVID-19 pandemic compared to before. The percentage of heavy alcohol drinkers and cigarette smokers also increased during the pandemic. Our findings corroborate results from several recent studies conducted in other countries. We also identified subgroups who were more vulnerable to unfavorable lifestyle changes during the pandemic. The most critical findings are for age and racial groups. Compared to Non-Hispanic Whites, Non-Hispanic Blacks and Hispanics were more likely to have undesired changes in multiple lifestyle behaviors, including exercise, screen time, fast food intake, and alcohol drinking; American Indians and those in the “other” racial category were more likely to decrease their exercise time and increase consumption of fast-food meals; while Asian Americans were less likely to increase alcohol drinking and cigarette smoking. The clustering of negative lifestyle changes among racial and ethnic minority groups, as revealed in this study, is consistent with findings from previous studies documenting the disproportionate exposure to and suffering from the COVID-19 pandemic by these groups [17,18,19,20]. Younger-aged adults were more likely to have undesired changes in exercise time, consumption of fast-food meals, alcohol drinking, and cigarette smoking compared with old adults. In addition, females had higher odds to increase screen time during the pandemic than otherwise similar males; married individuals were not more likely to decrease exercise time during the pandemic than those who were married or lived with partners; higher household income was associated with lower the odds of smoking during the pandemic; individuals with higher education levels had higher odds of having more undesired lifestyle changes during the pandemic, including less exercise time, more screen time, and more alcohol drinking Future lifestyle interventions (e.g., promoting more exercise, reducing screening time, reducing smoking, and alcohol drinking counseling) might become more effective if they can target these high-risk subgroups who are more likely to be disproportionately impacted and have more undesired lifestyle changes in a future pandemic or potential new COVID-19 outbreak.

It is no surprise that the pandemic resulted in a decline in exercise time and a rise in screen time. Indeed, 20.9% of individuals did not do any exercise during the pandemic (vs. 15.3% before) and 79.8% of individuals had screen time ≥4 h per day (vs. 59.5% before). The proportion of individuals who exercised ≥30 min per day also decreased to 45.4% during the pandemic (vs. 56.4% before). On average, the daily time spent on exercise decreased from 38.65 min in pre-pandemic to 32.06 min during the pandemic and screen time increased from 5.06 to 6.79 h. These findings are consistent with studies from other countries and populations [7,8,9,12]. For example, results from an international online survey showed that the COVID-19 had negative impact on all physical activity intensity levels (e.g., vigorous, moderate, walking) and the daily sitting time increased from 5.31 to 8.41 h [12]. In a study conducted in Italy, 56% of the study participants reported a reduced time for physical activity [8], where in our study 31.2% Americans reported a decrease in exercise time during the pandemic. Such changes toward a more sedentary lifestyle are likely caused by the restrictions of outdoor activities, in-gym exercise, and working from home. A study conducted in Italy found that the frequencies of running, walking, swimming, and in-gym training were all reduced during the COVID-pandemic compared to before the pandemic [9]. Given the health benefits of exercise and the harm of screen time (as an indicator of a sedentary lifestyle), such undesired changes are likely to lead to adverse influences on various health outcomes. Exercise and screen time are independently linked to multiple health outcomes in almost all age groups. Although the observed changes are small to modest, the cumulative impact of such negative changes over several months or longer could have extensive consequences.

We found that the COVID-19 pandemic had a negative impact on alcohol drinking and cigarette smoking among USA adults, which is in line with findings from studies conducted in the USA [14,15] and other countries such as Italy and Poland [8,11]. In our study, 9.0% of participants reported an increase in smoking during the pandemic, whereas this number was 29.5 and 45.2% in Italy and Poland, respectively [8,11]. Among smokers, the percentage of heavy smokers (i.e., ≥11 cigarettes/day) also increased to 7.9% during the pandemic (vs. 5.8% before). In our study population, 23.2% of participants had increased alcohol drinking and the percentage of heavy drinkers (i.e., ≥5 times/week) increased from 20.9% before the pandemic to 25.7% during the pandemic. Both alcohol drinking and smoking are risk factors for many health outcomes and have deleterious effects on the immune system, and therefore, can potentially increase the risk of SARS-CoV-2 infection and severity of the disease [21]. In addition, the evidence suggests that smoking is associated with a greater risk of adverse COVID-19 outcomes. Current and former smokers are 2.4 times more likely to need intensive care unit (ICU) support or die, compared with non-smokers [22]. The reasons for these negative changes are likely due to the stress and boredom of home confinement, as well as general life difficulties exacerbated by the COVID-19 pandemic [14]. Further studies are needed to understand the underlying causes and identify effective interventions in such emergency situations.

The observed decrease in fast food consumption is likely due to the stay-at-home order and the closure of fast-food restaurants during the pandemic. The average level of fast-meal consumption decreased from 1.41 times/week before the pandemic to 0.96 times/week during the pandemic. The percentage of individuals who reported consuming fast-food meals ≥3 times/week also decreased from 37.7% before the pandemic to 33.3% during the pandemic. Although the majority of study participants (77.4%) decreased or did not see a change in their fast-food meal consumption, there were still 22.6% that increased their fast-food meal consumption during the pandemic. In previous studies assessing different perspectives on eating, people reported more unhealthy dietary behaviors during the pandemic, including increased consumption of unhealthy foods (e.g., sweets), snacks, and the overall number of main meals [8,11,12]. Nevertheless, these data suggest that the COVID-19 pandemic and related regulations dramatically influence people’s eating behaviors and food consumption patterns. Although many health organizations have provided guidelines for healthy eating during the COVID-19 pandemic and disseminated information on how adequate nutrition can support the immune system [23,24], people from multiple countries did not put these recommendations into practice during the pandemic.

The strengths of this study include a large, nationally representative sample, which is based on a population of USA adults with adequate numbers from each racial/ethnic group, and the timing of the data collection. Our study sample included over 1000 Asian Americans, who are commonly underrepresented or excluded in other studies. The study survey was designed and carried out in fall 2020 at the peak of the USA COVID-19 pandemic; thus, the survey responses are timely and likely to reflect the acute impacts of the pandemic. This study has several noteworthy limitations. First, the survey data were only collected at the one-time point during the pandemic. Therefore, the changes in lifestyle behaviors may only reflect the peak of the pandemic. Second, we were only able to collect limited information for each lifestyle behavior due to the concerns regarding participant time and burden. For dietary intake, we only focused on the consumption of fast-food meals.

## 5. Conclusions

In this large study conducted during the peak of the COVID-19 pandemic in the USA, we found that regulations to restrict non-essential activities and the stay-at-home order during the pandemic had profound impacts on multiple lifestyle behaviors in American adults. We found a marked increase in sedentary behaviors, alcohol consumption, and cigarette smoking and a decline in exercise. These negative changes in lifestyle disproportionately impact racial and ethnic minorities who also bear a higher disease burden of COVID-19. Our findings corroborate results from several recent studies conducted in other countries. Collectively, the observed undesired lifestyle changes may have a negative impact on both physical and mental health outcomes. Future studies investigating whether such lifestyle changes persist as the pandemic continues and whether the quality of life and health well-being are subsequently affected, are warranted. Resources and supports that can help people maintaining healthy lifestyles during the pandemic are urgently needed, and different strategies may be needed for different sub-groups.

## Figures and Tables

**Figure 1 nutrients-13-03359-f001:**
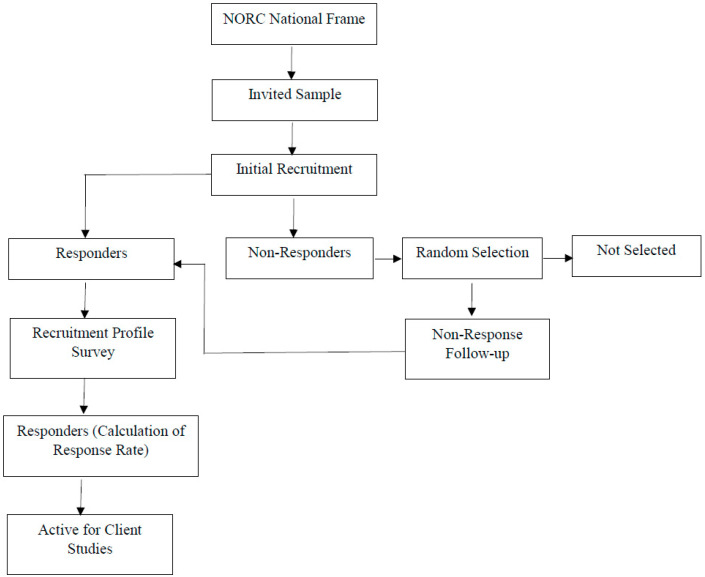
AmeriSpeak Panel Recruitment Methodology-Participants Selection Flowchart: The NORC national frame is an area probability sample of 3 million households based on the USA Postal Service Sequence File (covers approximately 97% of USA households) [16]. Abbreviations: NORC: National Opinion Research Center.

**Figure 2 nutrients-13-03359-f002:**
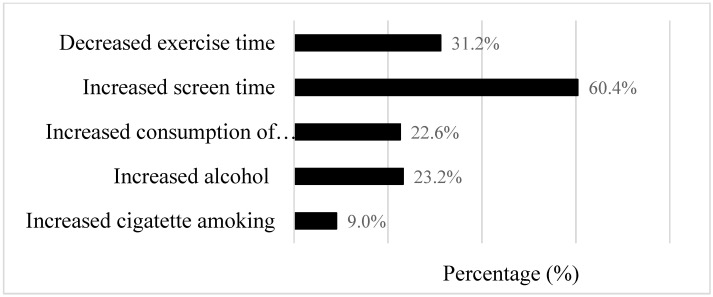
Bar chart for percentage (%) of undesired changes in 5 lifestyle behaviors during the COVID-19 pandemic. Changes in lifestyle behaviors were calculated (during pandemic–before pandemic) and categorized as “desired change” or “undesired change” for each behavior. The “undesired change” category refers to decreased exercise, increased screen time, increased numbers of fast-food meals, increased cigarettes smoking, or increased alcohol drinking during the pandemic as compared with before the pandemic, respectively, for each behavior.

**Table 1 nutrients-13-03359-t001:** Sociodemographic characteristics of HEAP study participants.

Characteristics	% (N)
Age groups	
18–29 years	20.5 (568)
30–44 years	25.5 (880)
45–59 years	24.2 (573)
≥60 years	29.8 (688)
Female	51.7 (1413)
Race/ethnicity	
Non-Hispanic White	61.3 (514)
Non-Hispanic Black	11.9 (590)
Asian and Pacific Islander	6.4 (1012)
Hispanic	16.7 (532)
American Indian or others	3.6 (58)
Education	
High school or less	38.0 (583)
Associates	27.7 (1073)
Bachelor’s or higher	34.3 (1053)
Marital status	
Married/living with a partner	58.2 (1444)
Widowed/divorced/separated	17.0 (404)
Never married	24.8 (861)
Annual household income	
<$25,000	18.8 (562)
$25,000–$49,999	22.5 (625)
≥$50,000	58.6 (1522)
Health insurance before the pandemic	
Private	52.7 (1480)
Uninsured	8.3 (222)
Medicare	21.3 (535)
Medicaid	17.7 (452)

Data were presented as weighted percentage, % (actual frequency, N).

**Table 2 nutrients-13-03359-t002:** Changes of lifestyle behaviors between before and during the COVID-19 pandemic.

Variables	Before the Pandemic	During the Pandemic	*p* Values
Total exercise time (min/day), mean (SE)	38.65 (2.02)	32.06 (1.66)	<0.001
Exercise time categories, % (N)			<0.001
0 min/day	15.3 (379)	20.9 (508)	
1–30 min/day	28.3 (722)	33.7 (888)	
≥30 min/day	56.4 (1517)	45.4 (1222)	
Total screen time (hours/day), mean (SE)	5.06 (0.13)	6.79 (0.14)	<0.001
Screen time categories, % (N)			<0.001
0 h/day	2.1 (52)	1.6 (47)	
<0 and <4 h/day	38.4 (988)	18.7 (474)	
≥4 h/day	59.5 (1576)	79.8 (2095)	
Fast-food meals intake (time/week), median (IQR)	1.41 (2.55)	0.96 (2.74)	<0.001
Fast-food meals intake categories, % (N)			<0.001
0 time/week	15.0 (400)	25.0 (663)	
1–2 time/week	47.3 (1142)	41.7 (1036)	
≥3 times/week	37.7 (1061)	33.3 (904)	
Alcohol drinking (drinks/week) among drinkers, median (IQR)	3.01 (5.13)	4.24 (6.92)	<0.001
Alcohol drinking categories, % (N)			<0.001
0 drink/week	50.6 (1409)	49.8 (1416)	
1–4 drinks/week	28.5 (806)	24.6 (687)	
≥5 drinks/week	20.9 (395)	25.7 (507)	
Cigarette smoking (times/day) among smokers, median (IQR)	9.29 (10.32)	9.80 (12.93)	<0.001
Cigarette smoking categories, % (N)			<0.001
Non-smokers	82.9 (2190)	82.4 (2185)	
1–5 cigarettes/day	5.8 (181)	3.9 (135)	
6–10 cigarettes/day	5.4 (143)	5.8 (149)	
≥11 cigarettes/day	5.8 (99)	7.9 (144)	

Data were presented as weighted mean (standard errors, SE) for continuous variables with normal distributions and median ((interquater range (IQR)) for continuous variables with non-normal distribution, and weighted percentage, % (actual frequency, N) for categorical variables. Before and during pandemic lifestyle difference were compared using Wilcoxon signed-rank test for continuous variables and Bhapkar’s test for categorical variables.

**Table 3 nutrients-13-03359-t003:** Multivariable associations between sociodemographic variables and the undesired changes of lifestyle before and during the COVID-19 pandemic.

Characteristics	Exercise Time		Screen Time		Fast-Food Meals Intake		Alcohol Drinking		Cigarette Smoking	
	OR (95% CI)	*p*	OR (95% CI)	*p*	OR (95% CI)	*p*	OR (95% CI)	*p*	OR (95% CI)	*p*
Age, years										
18–29	1.07 (0.65, 1.75)	0.79	1.25 (0.75, 2.10)	0.39	**2.24 (1.24, 4.05)**	**0.01**	**2.24 (1.23, 4.09)**	**0.01**	2.83 (0.95, 8.40)	0.06
30–44	**1.77 (1.13, 2.77)**	**0.01**	1.21 (0.78, 1.85)	0.40	**1.95 (1.16, 3.27)**	**0.01**	**2.34 (1.42, 3.86)**	**0.001**	**4.74 (1.80, 12.48)**	**0.002**
45–59	1.17 (0.72, 1.90)	0.52	0.82 (0.52, 1.28)	0.38	1.21 (0.71, 2.04)	0.49	1.33 (0.75, 2.36)	0.34	**3.37 (1.28, 8.85)**	**0.01**
≥60	Reference		Reference		Reference		Reference		Reference	
Sex										
Male	Reference		Reference		Reference		Reference		Reference	
Female	1.21 (0.91, 1.60)	0.20	**1.72 (1.29, 2.30)**	**<0.001**	0.95 (0.70, 1.29)	0.75	1.00 (0.72, 1.40)	1.00	1.31 (0.80, 2.15)	0.28
Race/ethnicity										
Non-Hispanic White	Reference		Reference		Reference		Reference		Reference	
Non-Hispanic Black	**1.64 (1.17, 2.28)**	**0.004**	1.30 (0.95, 1.79)	0.11	**1.49 (1.03, 2.17)**	**0.03**	**1.45 (1.02, 2.06)**	**0.04**	0.98 (0.55, 1.75)	0.95
Hispanic	**2.30 (1.64, 3.24)**	**<0.001**	**1.92 (1.35, 2.72)**	**<0.001**	**1.75 (1.21, 2.52)**	**0.003**	**1.59 (1.09, 2.33)**	**0.02**	0.73 (0.40, 1.34)	0.31
Asian/Pacific Islander	1.35 (1.00, 1.83)	0.05	1.03 (0.77, 1.37)	0.86	1.05 (0.74, 1.48)	0.79	**0.51 (0.35, 0.74)**	**0.001**	**0.53 (0.29, 0.97)**	**0.04**
American Indian or other	**3.56 (1.80, 7.07)**	**<0.001**	1.22 (0.58, 2.54)	0.60	**3.65 (1.79, 7.43)**	**<0.001**	2.04 (0.96, 4.33)	0.06	1.79 (0.62, 5.14)	0.28
Education										
High school or less	Reference		Reference		Reference		Reference		Reference	
Associate degree	**1.46 (1.03, 2.08)**	**0.03**	1.21 (0.85, 1.73)	0.30	1.11 (0.78, 1.56)	0.57	1.16 (0.76, 1.76)	0.49	0.96 (0.55, 1.67)	0.88
Bachelor degree or higher	**1.79 (1.20, 2.67)**	**0.004**	**1.90 (1.27, 2.85)**	**0.002**	1.05 (0.69, 1.60)	0.83	**1.78 (1.11, 2.87)**	**0.02**	0.52 (0.22, 1.24)	0.14
Marital Status, % (N)										
Married/Living with partner	Reference		Reference		Reference		Reference		Reference	
Widowed/Divorced/Separated	1.26 (0.80, 1.97)	0.32	0.82 (0.54, 1.26)	0.37	1.01 (0.65, 1.57)	0.96	1.08 (0.65, 1.81)	0.77	1.34 (0.62, 2.89)	0.45
Never married	**1.43 (1.02, 2.01)**	**0.04 ***	1.02 (0.69, 1.49)	0.93	0.76 (0.50, 1.16)	0.20	1.02 (0.69, 1.51)	0.94	1.00 (0.51, 1.96)	1.00
Annual household income										
<$25,000	Reference		Reference		Reference		Reference		Reference	
$25,000–$49,999	1.05 (0.70, 1.58)	0.82	1.04 (0.67, 1.60)	0.87	0.64 (0.41, 1.02)	0.06	1.00 (0.62, 1.61)	1.00	**0.37 (0.21, 0.67)**	**0.001**
≥$50,000	1.08 (0.70, 1.66)	0.74	1.36 (0.91, 2.06)	0.14	0.64 (0.41, 1.01)	0.05	1.08 (0.68, 1.71)	0.74	**0.22 (0.11, 0.43)**	**<0.001**
Health Insurance										
Private	Reference		Reference		Reference		Reference		Reference	
Uninsured	0.89 (0.53, 1.52)	0.68	1.37 (0.77, 2.44)	0.29	0.97 (0.55, 1.73)	0.92	0.74 (0.31, 1.76)	0.50	1.14 (0.47, 2.80)	0.77
Medicare	**1.63 (1.06, 2.52)**	**0.03**	1.06 (0.69, 1.62)	0.80	1.04 (0.63, 1.72)	0.88	0.61 (0.36, 1.03)	0.07	0.91 (0.34, 2.43)	0.85
Medicaid	1.51 (0.95, 2.39)	0.08	1.22 (0.78, 1.91)	0.39	1.38 (0.89, 2.15)	0.15	1.09 (0.68, 1.73)	0.73	**2.30 (1.03, 5.17)**	**0.04**

Abbreviations: OR: Odds ratio; 95% CI: 95% conference interval. Bold are statsitically significant results. * indicates *p* < 0.001.

## Data Availability

The data presented in this study are available on request from the corresponding author and the Center for Reducing Health Disparities at University of Nebraska Medical Center.

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
