# Peer review of "Changes of Exercise, Screen Time, Fast Food Consumption, Alcohol, and Cigarette Smoking during the COVID-19 Pandemic among Adults in the United States"

_nutrients, 2021, doi:10.3390/nu13103359_

Round 1

Reviewer 1 Report

This is a focused paper on the changes observed on several healthy lifestyles during COVID-19 pandemic of great interest for ‘Nutrients’. However, several major and minor issues are necessary to be attended in order to accept the manuscript:

Major

  • More information is needed on the different circumstances in which survey administration is applied and data gathering is developed. For instance, ‘How was the survey distributed for answering the different questions about the behaviours under study before and during the pandemic?’ ‘Was the survey applied after confinement periods?’ ‘Was the pandemic situation the same in the whole USA when participants answered the questions of the survey?’ To include a 'procedure section' is recommended.
  • I consider quite vague the measure of alcoholic drinks by number of drinks per week, since there are many different types and sizes of alcoholic drinks. At least, authors should justify this measure, and others, with an explanation and supported by previous studies that used the same measure.
  • Authors should check and provide more information in the ‘3.2. results subsection’ since data from the first paragraph, where they specify the values corresponding to the different lifestyle behaviour categories, do not match with data from the second paragraph and Figure 1, where they present global values by each category.
  • It is all right to identify, in the ‘discussion section’, the more vulnerable subgroups based on age and race, but it is also necessary to discuss the results from other subgroups based on education, marital status and household income.
  • A more complex and detailed ‘discussion’ is required when referring to screen time, alcohol drinking and cigarette smoking due to increasing and decreasing values in their different behavioural categories.
  • Authors make the following statement in the ‘discussion section’ (line 238): “The observed decrease in fast food consumption…”. However, at the end of the ‘3.2. Lifestyle behaviors…’ section, authors indicate “22.6% increased consumption of fast-food meals” (line 157). This is a contradiction that should be attended.

Minor

  • The first paragraph of the manuscript contains information on the pandemic and measures for mitigating COVID-19. Although most of this information is known by general population, ‘Nutrients’ is a scientific journal and references should support the knowledge about the characteristics of the disease and the context created by its social consequences.
  • Authors should reconsider the writing of the following sentence (208-210 lines): “Future interventions might become more targeted and effective in addressing health disparities associated with COVID-19 if they can disproportionately engage these high-risk groups who experienced significant negative changes in lifestyles during the pandemic”. I recommend to substitute ‘future interventions’ by ‘social measures’ or similar words and also refer to future pandemics or potential new COVID-19 outbreaks (and not only the present pandemic).
  • Avoid statements like this in the ‘conclusion section’: “Our findings corroborate results from several recent studies conducted in 267 other countries.” (lines 267-268), since they are more appropriate for a ‘discussion section’.
  • If authors mention “interdisciplinary interventions” instead of social measures or regulations in the ‘conclusion section’, they should specify which type of interventions or add a few examples in order to help readers to understand future actions among vulnerable groups.

Author Response

Reviewer 1

This is a focused paper on the changes observed on several healthy lifestyles during COVID-19 pandemic of great interest for ‘Nutrients’. However, several major and minor issues are necessary to be attended in order to accept the manuscript:

Major

  1. More information is needed on the different circumstances in which survey administration is applied and data gathering is developed. For instance, ‘How was the survey distributed for answering the different questions about the behaviours under study before and during the pandemic?’ ‘Was the survey applied after confinement periods?’ ‘Was the pandemic situation the same in the whole USA when participants answered the questions of the survey?’ To include a 'procedure section' is recommended.

RE: Thanks for your comments. We have added details about the participants recruitment and selection in the revised manuscript, with a new figure (Figure 1, page 3) to show the participants selection procedure and methodology. We clarified that the survey was conducted between October and November 2021 during the confinement period in the US. We also clarified that we asked participants to report the average frequency of each behavior during the pandemic and recall what they did on average before the pandemic (page 2, line 99; Page 3, line 100-113). 

  1. I consider quite vague the measure of alcoholic drinks by number of drinks per week, since there are many different types and sizes of alcoholic drinks. At least, authors should justify this measure, and others, with an explanation and supported by previous studies that used the same measure.

RE: We agree with the review that the measure of the alcohol drink in HEAP study is not comprehensive. We only asked for the average frequency of alcohol drinking before and during the pandemic. We have clarified that one drink is equivalent to a 12-ounce beer, a 5-ounce glass of wine, or a drink with one shot of liquor (page 4, line 130-131) which is the question used in the US Behavioral Risk Factor Surveillance System (BRFSS) (https://www.cdc.gov/brfss/index.html).  We acknowledged that it is a study limitation that we were not able to measure each behavior comprehensively (page 10, line 346).  The following studies should have more comprehensive measured of each behavior and their impact on health outcomes.

  1. Authors should check and provide more information in the ‘3.2. results subsection’ since data from the first paragraph, where they specify the values corresponding to the different lifestyle behaviour categories, do not match with data from the second paragraph and Figure 1, where they present global values by each category.

RE: Thanks for your comments. In the Results section 3.2., the first paragraph describes results in Table 2, which are the distributions (i.e., mean or median, as well as the percentage on each category) of the 5 lifestyle behaviors before and after the pandemic separately in study participants. For example, the average level of exercise time and the % of 0, 0-30, and ≥30 min/day. The second paragraph described the results in Figure 2, which we calculated the changes of 5 lifestyle behaviors (i.e., values during pandemic – values before pandemic) for each participant and then categorized their changes into desired changes or undesired changes. For example, the first bar in the figure 2 indicated that 31.2% of study participants decreased their exercise time during the pandemic compared with before pandemic. We have clarified such difference in the revised manuscript (page 5, line 172, 180-181, 184-186, 189-191; page 6, line 201-204) and added more information on the legend of Figure 2 (page 6, line 210).

  1. It is all right to identify, in the ‘discussion section’, the more vulnerable subgroups based on age and race, but it is also necessary to discuss the results from other subgroups based on education, marital status and household income.

RE: Thanks for your comments. We have added the results from other subgroups based on education, marital status, and income in the discussion session (page 9, line 270-276).

  1. A more complex and detailed ‘discussion’ is required when referring to screen time, alcohol drinking and cigarette smoking due to increasing and decreasing values in their different behavioural categories.

RE: Thanks for your comments. We have added the more detailed discussions according to your suggestions (page 9, line 290-292, 293-302; page 10, 311-313, 315-316).

  1. Authors make the following statement in the ‘discussion section’ (line 238): “The observed decrease in fast food consumption…”. However, at the end of the ‘3.2. Lifestyle behaviors…’ section, authors indicate “22.6% increased consumption of fast-food meals” (line 157). This is a contradiction that should be attended.

RE: Thanks for your comments. Based on the results in Table 2, the average level (fast-food consumption frequency per week) decreased during the pandemic compared to the before pandemic (1.41 vs. 0.96). The percentage of individuals who reported consuming fast-food meals ≥3 times/week also decreased from 37.7% before the pandemic to 33.3% during the pandemic. However, there are 22.6% of study participants who increased their fast-food meal consumption during the pandemic. In other words, majority of study participants (77.4%) decreased or unchanged their fast-food meal consumption. We have added more information to clarify such differences (page 5, line 180-182; page 6, 201-204; page 10, 327-331).

Minor

  1. The first paragraph of the manuscript contains information on the pandemic and measures for mitigating COVID-19. Although most of this information is known by general population, ‘Nutrients’ is a scientific journal and references should support the knowledge about the characteristics of the disease and the context created by its social consequences.

RE: Thanks for your comments. We have added the information regarding the COVID-19 and SARS-CoV-2 virus in the revised manuscript (page 2, line 58-70). 

  1. Authors should reconsider the writing of the following sentence (208-210 lines): “Future interventions might become more targeted and effective in addressing health disparities associated with COVID-19 if they can disproportionately engage these high-risk groups who experienced significant negative changes in lifestyles during the pandemic”. I recommend to substitute ‘future interventions’ by ‘social measures’ or similar words and also refer to future pandemics or potential new COVID-19 outbreaks (and not only the present pandemic).

RE: Thanks for your comment. We think “interventions” are more appropriate in here but agreed to modify this sentence as “Future lifestyle interventions (e.g. promoting more exercise, reducing screening time, reducing smoking and alcohol drinking counseling) might become more effective if they can target on these high-risk subgroups who are more likely to be disproportionately impacted and have more undesired lifestyle changes in future pandemic or potential new COVID-19 outbreak” (page 9, line 279-284).  

  1. Avoid statements like this in the ‘conclusion section’: “Our findings corroborate results from several recent studies conducted in 267 other countries.” (lines 267-268), since they are more appropriate for a ‘discussion section’.

RE: Thanks for your comment. We moved this statement to the discussion section (page 9, line 257).

  1. If authors mention “interdisciplinary interventions” instead of social measures or regulations in the ‘conclusion section’, they should specify which type of interventions or add a few examples in order to help readers to understand future actions among vulnerable groups.

RE: Thanks for your comment. We believe our results indicated that national-wide lock-down regulations could have profound impact on people’s behaviors and they might influence the sub-populations in different scopes. For example, interventions to reduce screen time should target on females, but not on males. Interventions to reduce alcohol drinking may need to target on Blacks and Hispanics, but not on Asians. We have added the type of intervention (page 9, line 279-284) in the revised manuscript as described in the #7 comment.   We also revise the last sentence in the conclusion section as “Resources and supports that can help people maintaining healthy lifestyles during the pandemic are urgently needed, and different strategies may be needed for different sub-groups” (page 11, line 362-364).

Reviewer 2 Report

Dear authors, first of all I would like to thank you for the effort and time dedicated to the preparation and submission of this manuscript, however, However, my decision was to reconsider accepting the manuscript after an important revision, it is appropriate to point out some suggestions for improvement in the different sections of the article.
The study focuses on a relevant research area considering the global impact of the pandemic, such as COVID-19. A revision of the manuscript is required to increase the quality of the scientific and academic writing style. The authors should improve the Introduction, it is suggested that they do a bibliographic review of the studies published in the last 5-10 years and that they incorporate a greater number of citations to give more solidity to their arguments.
In its current form, the materials and methods section is neither clear nor sufficient. You should review following a clearer structure of study design, participants, procedures, statistical analysis…. In the Method section, it would be necessary for the authors to indicate: the type of study (quantitative / qualitative; descriptive, observational, cross-sectional ... etc. Data Protection, etc ...)
The Presentation of the Results would be much better understood if the Authors placed The Tables in that section and none of the United Nations As Annex After the references
The Discussion should be a space where the authors discuss the results found in their work with other equal or similar studies. Although the study is novel, the authors should include studies published in the last 5-10 years to give more solidity to their arguments.
References: The list of references is not in accordance with the journal guidelines. Please check accordingly. (Check everything by modifying the punctuation marks between the authors and indicate the year of publication in bold) Indicate the existence or absence of conflicts of interest.

Author Response

  1. Dear authors, first of all I would like to thank you for the effort and time dedicated to the preparation and submission of this manuscript, however, However, my decision was to reconsider accepting the manuscript after an important revision, it is appropriate to point out some suggestions for improvement in the different sections of the article.
    The study focuses on a relevant research area considering the global impact of the pandemic, such as COVID-19. A revision of the manuscript is required to increase the quality of the scientific and academic writing style. The authors should improve the Introduction, it is suggested that they do a bibliographic review of the studies published in the last 5-10 years and that they incorporate a greater number of citations to give more solidity to their arguments.

RE: Thanks for your comments. We have added some studies on health behaviors’ changes during the 2003 SARS pandemic (page 2, line 68-70).  

  1. In its current form, the materials and methods section is neither clear nor sufficient. You should review following a clearer structure of study design, participants, procedures, statistical analysis…. In the Method section, it would be necessary for the authors to indicate: the type of study (quantitative / qualitative; descriptive, observational, cross-sectional ... etc. Data Protection, etc ...)

RE: Thanks for your comments. We have added more information in the Methods section, including the cross-sectional design of the study, the detailed sampling selection methods and procedures, and the IRB information. We also added new figure (Figure 1) and reference (#16) to show the participants selection procedure (page 3).

  1. The Presentation of the Results would be much better understood if the Authors placed The Tables in that section and none of the United Nations As Annex After the references

RE: Thanks for your comments. We have placed the tables and figures in the appropriate sections.

  1. The Discussion should be a space where the authors discuss the results found in their work with other equal or similar studies. Although the study is novel, the authors should include studies published in the last 5-10 years to give more solidity to their arguments.

RE: Thanks for your comments. We have added more discussions to compare our results to the similar studies in the revised manuscript suggestions (page 9, line 290-292, 293-302; page 10, 311-313, 315-316).

To the best of our knowledge and a recent publication (Piret J, Boivin G. Pandemics Throughout History. Front Microbiol. 2021 Jan 15. PMID: 33584597; PMCID: PMC7874133.), there were several pandemics of infectious diseases in human history, such as plague, cholera, and influenza. Because of little scientific evidence we could retrieve regarding the specific topic of lifestyle changes during infectious disease pandemics, we have tried hard, as you suggested, and added the previous studies reported lifestyle changes during the SARS pandemic in the introduction section. In the past two decades, the only one pandemic caused by coronavirus is the current COVID-19 that results in lock-down regulations in infected countries and have profound impact in human daily life. We have included these studies in the revised manuscript (page 2, line 68-70).

  1. References: The list of references is not in accordance with the journal guidelines. Please check accordingly. (Check everything by modifying the punctuation marks between the authors and indicate the year of publication in bold)

RE: Thanks for your comments. We have changed the reference style according to the journal guidelines.

  1. Indicate the existence or absence of conflicts of interest.

RE: Thanks for your comments. We have added the conflict of interest statement in the revised manuscript (page 11, line 375-376).

Round 2

Reviewer 1 Report

Authors have substantially well clarified all the issues I recommended in the first review of the manuscript. However, there are still few minor issues linked to the writing that should be amended before publication of the paper in ‘Nutrients’. I am particularly referring to:

  • Many mistakes or unclear issues in the writing of some words in the added text. For instance: server, in peopled lived in, methodlogy, flowchat, Novmber, cross-section study, paterners, Intaly, unchaged, … And a sentence or part of it that is repeated: “The survey was delivered in both English and Spanish through Web or telephone.”. See lines 108-109 and 112-113.
  • Modified and added sentence in lines 327-331 are clearer explained in the notes addressed to me that in the manuscript. Please, consider revising this part for facilitating readers’ understanding.

Author Response

Authors have substantially well clarified all the issues I recommended in the first review of the manuscript. However, there are still few minor issues linked to the writing that should be amended before publication of the paper in ‘Nutrients’. I am particularly referring to:

  • Many mistakes or unclear issues in the writing of some words in the added text. For instance: server, in peopled lived in, methodlogy, flowchat, Novmber, cross-section study, paterners, Intaly, unchaged, … And a sentence or part of it that is repeated: “The survey was delivered in both English and Spanish through Web or telephone.”. See lines 108-109 and 112-113.

RE: Thanks very for your comments. We apology for these typos and have corrected them in the revised manuscript. We also removed the repeated sentence on page 3, line 112-113.

  • Modified and added sentence in lines 327-331 are clearer explained in the notes addressed to me that in the manuscript. Please, consider revising this part for facilitating readers’ understanding.

RE: Thanks very for this comment. We have revised this sentence in the revised manuscript as “The average level of fast-meal consumption decreased from 1.41 times/week before the pandemic to 09.6 times/week during the pandemic. The percentage of individuals who reported consuming fast-food meals ≥3 times/week also decreased from 37.7% before the pandemic to 33.3% during the pandemic. Although majority of study participants (77.4%) decreased or unchanged their fast-food meal consumption, there were still 22.6% of them increased their fast-food meal consumption during the pandemic.”

Reviewer 2 Report

Dear authors, thank you for attending to the suggestions made in the previous review. However, they should consider performing the following minor checks: 1.- In the introduction they should separate in a final paragraph, the objective of the study, so that it is clearly distinguished from what is the foundation of the investigation.
2.- They must write the bibliographic references according to the regulations of the journal. Journal references are written as follows:
For example, for magazine articles: 1. Author 1, AB; Author 2, CD Title of the article. Magazine abbreviated name Year, volume, page range.
(To see how to write the references of other types of documents, go to the instructions for authors of the journal)
Authors should review all references emphasizing the following aspects • Authors must be separated by; • There is no point between the authors and the title of the article •
The year of publication is highlighted in black font

Greetings

Author Response

Dear authors, thank you for attending to the suggestions made in the previous review. However, they should consider performing the following minor checks:

1.- In the introduction they should separate in a final paragraph, the objective of the study, so that it is clearly distinguished from what is the foundation of the investigation.

RE: Thanks for your comment. We have separated study objective in a new paragraph.

2.- They must write the bibliographic references according to the regulations of the journal. Journal references are written as follows:
For example, for magazine articles: 1. Author 1, AB; Author 2, CD Title of the article. Magazine abbreviated name Year, volume, page range.
(To see how to write the references of other types of documents, go to the instructions for authors of the journal)
Authors should review all references emphasizing the following aspects • Authors must be separated by; • There is no point between the authors and the title of the article •The year of publication is highlighted in black font

RE: Thanks for bring us to the details of the reference format. We have changed the cited references according to the journal’s requirements (i.e., ACS style).